# A 3D Miniaturized Glass Magnetic-Active Centrifugal Micropump Fabricated by SLE Process and Laser Welding

**DOI:** 10.3390/mi13081331

**Published:** 2022-08-17

**Authors:** Jeongtae Kim, Sungil Kim, Jiyeon Choi, Chiwan Koo

**Affiliations:** 1Department of Electronic Engineering, Hanbat National University, Daejeon 34158, Korea; 2Department of Laser and Electron Beam Technologies, Korea Institute of Machinery and Materials, Daejeon 34103, Korea; 3Currently with Corning Technology Center Korea, Corning Precision Materials Co., Ltd., Asan 31454, Korea

**Keywords:** centrifugal pump, glass micropump, glass microfluidics, laser welding, SLE process

## Abstract

A miniaturized pump to manipulate liquid flow in microchannels is the key component of microfluidic devices. Many researchers have demonstrated active microfluidic pumps, but most of them still required additional large peripherals to operate their micropumps. In addition, those micropumps were made of polymer materials so that their application may be limited to a variety of fields that require harsh conditions at high pressures and temperatures or organic solvents and acid/base. In this work, we present a 3D miniaturized magnetic-driven glass centrifugal pump for microfluidic devices. The pump consists of a volute structure and a 3D impeller integrated with two magnet disks of Φ1 mm. The 3D pump structure was 13 mm × 10.5 mm × 3 mm, and it was monolithically fabricated in a fused silica sheet by selective laser-induced etching (SLE) technology using a femtosecond laser. The pump operation requires only one motor rotating two magnets. It was Φ42 mm × 54 mm and powered by a battery. To align the shaft of the motor to the center of the 3D glass pump chip, a housing containing the motor and the chip was fabricated, and the overall size of the proposed micropump device was 95 mm × 70 mm × 75 mm. Compared with other miniaturized pumps, ours was more compact and portable. The output pressure of the fabricated micropump was between 215 Pa and 3104 Pa, and the volumetric flow rate range was 0.55 mL/min and 7.88 mL/min. The relationship between the motor RPM and the impeller RPM was analyzed, and the flow rate was able to be controlled by the RPM. With its portability, the proposed pump can be applied to produce an integrated and portable microfluidic device for point-of-care analysis.

## 1. Introduction

Microfluidic devices and lab-on-a-chip technologies have great potential due to their small form factor properties for point-of-care (POC) devices, such as transportable, portable, and handheld instruments [1,2,3]. However, although microfluidic devices are portable, commercial and bulky pumping systems are still utilized to manipulate fluid in microfluidic channels. Therefore, conventional pumps are required to be miniaturized for being used in complete POC devices. To miniaturize and simplify the pumping system, numerous researchers have demonstrated microscale pumps using passive or active methods [4,5,6,7,8,9,10]. Passive manipulation strategies usually rely on the geometry and design of the microfluidic channels [11,12,13]. Not only physical methods but also several chemical methods have been developed. Diffusion–osmotic methods generate the flow by an osmotic pressure difference [14,15], and the method of self-oscillation of polymers directly converts chemical energy to mechanical energy by chains of chemical reaction [16,17]. Although these passive methods can easily make the fluidic flow without any external force, precise fluid control is difficult. On the other hand, active methods using external forces such as electrical, magnetic, acoustic, electrohydrodynamic, and thermodynamic actuation can handle liquid on demand. For example, Ozcelik et al. have introduced a simple acoustofluidic pump [18]. They fabricated it by assembling glass capillaries with two 3D-printed adaptors. It was compact as 55 mm × 22 mm × 3 mm, and its flow rate was 0.012 mL/min. Wang et al. have reported a micro heat pipe device for nanofluidics [19]. Their device generates continuous flow by vapor pressure and condensation of the vapor. However, the integration of active micropumps in microfluidic devices often restricted the pumping performance by a relatively low flow rate.

To achieve the goal of a micropump with a high flow rate, various types of miniaturized pumps are actively being researched. Wang et al. have suggested a low-cost and easy assembly portable pump using a commercially available microblower [20]. Their pump system showed reasonable performance in flow rate up to 128 mL/min. Although the micro-blower size was suitable for a portable pump system, an accessory structure such as a liquid reservoir was required to inject liquid into the microfluidic device. Park et al. have reported a finger-actuated microfluidic pump and valve device made of polydimethylsiloxane (PDMS) [21]. The pump and valve can generate liquid flow by the press deflection of the PDMS membrane with a finger. Although their device could generate liquid flow and control flow direction without an external device, it could not control flow rate and supply liquid continuously. Al-Halhouli et al. have demonstrated a centrifugal pump integrated with a microfluidic compact disk (CD) platform [22]. They implanted a magnet into the pump and rotated the pump impeller by spinning another magnet outside. The flow rate of their pump reached up to about 10 mL/min, and the liquid was migrated successfully from one reservoir to another one. However, multiple motors were required to operate the microfluidic function of the CD, and it leads to the enlargement of the size of the microfluidic device. Another way to miniaturize a macroscopic pump is a 3D printing method. Similar to the centrifugal pump integrated with the CD platform, Joswig et al. have reported a miniature 3D-printed centrifugal pump [23]. They also used an implanted magnet to rotate the impeller but did not use a motor. They generated the magnetic field using four coils and a custom circuit board to reduce the overall size of the pump device. Its size was considerably small, at 28 mm × 30 mm × 24 mm, but additional apparatus such as a DC power supply and a control board were required to operate the pump. Mao et al. have reported a stacked electrohydrodynamic pump using dielectric and functional fluids. It is only 113 mm × 20 mm × 3 mm but requires an external high-voltage source [24].

Most micropumps are made of polymer, and they cannot be used for various applications due to their poor resistivity to chemicals, such as solvent and acid/base [25,26]. Glass is well-known for its high chemical and thermal resistance and is a good alternative material for polymer. Although its properties make glass suitable for microfluidic devices in various fields, it also makes the fabrication of glass microfluidic devices difficult. Fortunately, in recent years, many researchers have demonstrated glass micromachining technology using an ultrafast laser, which is called selective laser-induced etching (SLE) [27]. SLE is a two-step hybrid process consisting of ultrafast laser modification and wet etching for the fabrication of three-dimensional (3D) glass microfluidic devices and is one of the technologies that efficiently overcomes the limitations of the existing 2D MEMS processes [28,29,30,31,32]. Due to some nonlinear optical effects, including multiphoton absorption and local plasma excitation, the highly focused ultrashort pulse laser beam undergoes complex chemical and physical properties modifications of material. For this reason, it is possible to selectively remove the modified area without masking [33,34,35]. We suggested guidelines for the efficient fabrication of microstructures by irradiating laser parameters with up to 333 times faster etching rate inside the glass [36]. In addition, we have recently succeeded in downscaling microvalve and micromixer, realizing the manufacture of high-performance glass biochips [37,38]. In addition, ultrafast laser micro welding technology provides high bonding strength through simple and fast direct bonding of glass to glass without a chemically adhesive layer through high repetition rate pulse control [39,40,41,42].

To overcome drawbacks of the previously reported micropumps, we present a 3D miniaturized glass centrifugal pump using the electromagnetic actuation method. The proposed glass micropump has three advantages. First, the micropump is made of glass. Glass microfluidic devices exhibit high mechanical strength and chemical stability and are superior to organic polymers such as PDMS, which is a popular material for microfluidic devices. The proposed pump is able to pump various chemical solutions. Second, it requires only one small motor to operate the proposed micropump without any additional peripherals. The impeller inside the micropump had magnets so that it could be rotated by the rotation of external magnets mounted on a motor. A small and portable motor operated by a battery is commercially available. Third, all elements within the device including the impeller, rotating shaft, and volute structure are monolithically fabricated in a fused silica substrate using SLE. Thus, complicated assembly processes are eliminated, and the high durability of the built-in glass parts can be obtained in monolithic fabrication.

In this work, we describe a novel 3D miniaturized centrifugal pump. To obtain a high pumping rate, a 3D impeller and volute structure were designed and fabricated monolithically using the SLE technique, and a portable motor with two magnets and a controller circuit to rotate the impeller were manufactured. We also evaluated the performance of our pump by pumping water and measuring the weight and compared our results with that of conventional micropumps.

## 2. Materials and Methods

### 2.1. Design of a 3D Glass Magnetic-Active Centrifugal Micropump for Microfluidic Applications

A glass micropump was designed based on the shape of a conventional centrifugal pump with a volute structure. Our pump is composed of three glass parts, as shown in Figure 1A: an upper glass, an impeller, and a cover glass. The most important part is the upper glass, because it has all the functional structures of the centrifugal pump. It has a chamber shape with an inlet and an outlet hole and a shaft. The chamber consists of three layers. The first layer has an inlet to draw liquid into the chamber and the shaft for an impeller at the center of the chamber. The first layer is circular in shape, and its diameter and height are 3.7 mm and 0.34 mm, respectively. The inlet hole is located at the center of the micropump, and its diameter and height are 2 mm and 0.22 mm, respectively. The shaft is suspended from four beams towards the second layer, and its dimensions are 0.64 mm and 0.52 mm in diameter and height, respectively.

The second layer is a volute shape, which has a height of 0.83 mm and enlarged diameters from 7.5 mm to 9 mm. The outlet is connected with the second layer by a straight channel. Its height is 1.41 mm, and its diameter is 1.5 mm. The third layer having the impeller is a cylindrical shape with a 7.1 mm diameter and a 0.75 mm height. The impeller structure is designed as a closed type with six blades and has two insert holes for circular magnets. The diameter and height of the impeller are 7 mm and 2.1 mm, respectively. The shaft hole is designed as a hollow cylindrical shape at the center of the impeller, and its dimensions are 0.7 mm, 1.1 mm, and 0.4 mm in inner diameter, outer diameter, and height, respectively.

### 2.2. Fabrication Methods

The main procedure of device fabrication via SLE and bonding consists of three steps, as shown in Figure 2. A 25 mm × 25 mm × 3 mm fused silica (JMC glass, Gyeonggi, Korea) substrate was used to fabrication of the 3D glass micropump. An Yb-doped femtosecond laser amplifier (Satsuma HP2, Amplitude systems, Pessac, France) with a center wavelength of 1030 nm was used for laser modification (laser writing). The modification parameters for high etching selectivity conditions, pulse width: 1 ps, and pulse repetition rate: 500 kHz, controlled with a pulse energy of 400 nJ [36,37,38]. The laser beam was focused on an objective scanner system that combines a 2-axis (X, Y) Galvano scanner (DynAXIS, ScanLab, Puchheim, Germany) and a 50X objective lens (NA 0.42, Mitutoyo, Kawasaki, Japan). Additionally, the slicing and hatching scales of the tool paths extracted from the 3D design file (STL file) were set to 15 µm. The modified glass substrate was immersed in a potassium hydroxide (KOH) container at a concentration of 8 mol/L and then etched in a 90 °C ultrasonic bath for 22 h. After assembling a neodymium magnet on the impeller of the etched 3D micropump, a 1 mm thick cover glass was pre-bonding with optical contact. After replacing the 20X objective lens (NA 0.42, Mitutoyo, Kawasaki, Japan), a welding line was formed along the pump structure. At this time, the pulse repetition rate and pulse energy were controlled to 2 MHz and 10 μJ, respectively [42].

### 2.3. Operation Principle and Performance Evaluation of Micropump

Figure 3A shows an operation principle of our micropump. There are two small magnets (internal magnets) inserted in the impeller of the micropump and two large magnet disks (external magnets) on an external motor. The internal magnets and the external magnets have same magnetic polarity each other. Because of their same magnetic polarity, the internal magnets are always repelled by the external magnets. When the external magnets rotate by the motor, the internal magnets also rotate by the repulsion force. The impeller is continuously rotated until the external magnets are stopped.

Figure 3B shows the pump performance evaluation setup. It is composed of a micropump chip, a motor control system, and liquid reservoirs. The micropump chip was located above the motor control system, and the motor was aligned to the micropump chip so that the shaft of the motor is coaxial with the shaft of the micropump. For a more accurate position, a motor system housing was with slots for the micropump chip and the motor was manufactured by a 3D printer. The motor controller was also mounted on the bottom of the motor system housing. The revolution speed of the motor was controlled by an on-board potentiometer of the motor controller. To pump liquid, the micropump was connected to a liquid reservoir containing water with an 8 mm (OD) tube, and the outlet was connected to an empty reservoir. To evaluate the performance of the micropump, the volumetric flow rate through the outlet was measured while increasing the motor revolution speed with a 500 RPM step. The pumping time was 30 s, and the gathered liquid from the outlet was measured using a precision electronic scale (AS 82/220.R2, Radwag, Radom, Poland).

## 3. Results and Discussion

### 3.1. Fabrication Results

The upper glass and an impeller were monolithically prototyped in a glass chip using SLE and a cover glass was bonded to enclose the chamber with the impeller. Figure 4A shows 3D digital microscopy images (KH-8700, Hirox, Tokyo, Japan). All structures of glass micropump were successfully fabricated as the design dimension in the XY plane. The micropump measures 11.4 mm × 9.72 mm × 2.94 mm. The inlet and outlet holes were 200 μm and 1.8 mm in height, respectively. The impeller measures 2.00 mm in height. Two holes to implant the magnets measure 1.13 mm in height, respectively. The shaft for the impeller was fabricated in a cylindrical shape and mounted on four curved beams as in the design. The height of the shaft was 525 μm. Other dimensions are shown in the figure.

### 3.2. Performance Evaluation of 3D Glass Magnetic-Active Centrifugal Micropump

The performance evaluation setup of the micropump system is shown in Figure 4B. The inlet of the glass micropump was connected with the reservoir with the suction tube and the outlet was connected with the conical tube with the discharging tube. To fix the tubes on the inlet and outlet, an epoxy bond was applied at the interface between the glass chip and tubes. The external magnets were equipped at the shaft of the motor by enclosing with the 3D-printed housing. To rotate the external magnets smoothly, there was a 1 mm clearance between the chip loading platform and the housing of external magnets. The overall size of the micropump system was measured at 95 mm × 70 mm × 75 mm.

When the control signal was applied to the motor, the external magnet was spun so that the rotating magnetic field was generated. As a result, the impeller inside the glass micropump was rotated through the spinning magnetic field, as shown in Figure 5A. To show the rotation of the impeller clearly, a real-time video of the rotating impeller is shown in Appendix A. To observe liquid suction and discharging of our glass micropump, it was operated by rotating the external magnet at 500 RPM. At this speed, the liquid was not discharged despite the rotation of the impeller. To find a proper RPM to pump water, the motor RPM was increased with a 500 RPM step. When the RPM reached near 3000 RPM, suction and discharging of the liquid were observed in the inlet and the outlet (Figure 5B). A real-time suction and discharging video of blue-dyed water by operating pump is shown in Appendix A. To find a starting point of the micropump, the RPM was decreased from 3000 RPM with a step of 100 RPM. The discharging of the liquid was stopped under 2700 RPM. If the RPM increased to 2700 RPM, suction and discharging of the liquid were observed again.

To calculate the volumetric flow rate at 2700 RPM, gathered liquid in the conical tube for 30 s was measured using the precision electrical scale. The weight was 0.27 g, and the density of the water was 0.998 at 25 °C, so the flow rate at 2700 RPM was 0.54 mL/min.

The output pressure of the pump can be evaluated by measuring the heights of the pumped fluid, which is the head of the pump. To show the pressure performance of our micropump, the height of the water was measured five times by increasing the motor RPM. Finally, the measured head height was converted to the pressure. At 2700 RPM, the output head height was 2.2 cm, and the pressure was 215 Pa. The pressure of our micropump was increased from 570 Pa at 3000 RPM to 3104 Pa at 6000 RPM. It means that the maximum output pressure is about 3104 Pa.

To evaluate the flow rate depending on the rotation speed of the external magnet, the flow rate was measured by gathering the liquid in the conical tube, increasing the RPM of the motor from 3000 RPM to 6000 RPM as a 500 RPM step. The measurement was repeated five times. Figure 6 shows the graph of the flow rate and its relation to RPM. The flow rates from 3000 RPM to 6000 RPM were 0.97 mL/min, 1.68 mL/min, 2.10 mL/min, 3.33, 5.01, 5.89, and 7.88 mL/min, respectively. To analyze the linearity of our pumping rate depending on the motor RPM, the flow rate data were fitted using a least square method. In Figure 6A, the flow rate was non-linear as 0.99 in R-square for a third-order polynomial curve in all ranges from 3000 RPM to 6000 RPM. It is more intuitive and easier to control the pumping system when the flow rate is linear to the RPM. Thus, the flow rate was investigated by dividing the RPM ranges into two ranges to obtain a linear response of the flow rate to the RPM. In Figure 6B, the increase in the flow rate was higher when the RPM of the motor was changed from 4000 to 4500, but the flow rate from 3000 RPM to 4500 RPM was linear as 0.96 in R-square. The flow rate from 4500 RPM to 6000 RPM was increased more linearly as R-square is 0.98 than one from 3000 RPM to 4500 RPM, as shown in Figure 6C.

### 3.3. Rotation Speed Estimation of 3D Glass Magnetic-Active Centrifugal Micropump

The flow rate of the centrifugal pump is theoretically proportional to impeller RPM. Because the flow rate was not proportional to the motor RPM, it implies that the rotation speed of the impeller did not correspond with that of the motor. It is the reason why the impeller was indirectly connected with the motor by magnetic force, in our device. To estimate the impeller rotation speed, the measured volumetric rates were compared with the theoretical volumetric rate. The theoretical flow rate can be calculated by the following [42]:(1)$$q_{t}=n\cdot RPM_{m}\cdot V$$
where *q_t_* is the theoretical flow rate, *n* is the number of pockets between two impeller blades, *RPM_m_* is the RPM of the motor, and *V* is the volume of one pocket. By Bernoulli’s equation, the flow rates at zero and non-zero heads can be represented in the following equation:(2)$$q_{tx}=\rho \cdot A\cdot \sqrt{{\left(\frac{q_{t0}}{\rho A}\right)}^{2}+2\cdot g\cdot h}$$
where *q_tx_* is a flow rate at a specific head, *ρ* is a density of a fluid, *A* is the cross-sectional area of the channel, *q_t0_* is a flow rate at zero head, *g* is the gravitational acceleration, and *h* is a head height. The estimated impeller RPM, *RPM_i_*, was calculated by the following equation:(3)$$RPM_{i}=RPM_{m}\cdot \frac{q_{c}}{q_{t}}$$
where *q_c_* is the measured flow rate.

When comparing the measured flow rate with the theoretical flow rate, the estimated impeller RPMs were about 24, 43, 53, 83, 120, 144, and 204 while increasing the motor RPM from 3000 to 6000 RPM with a step of 500 RPM. The motor RPM was not as linear as the flow rate, but the flow rate-the estimated RPM curve was perfectly linear, as shown in Figure 7A. It means that the flow rate of our pump can be controlled linearly using the motor RPM-the estimated RPM fitting curve. Using the RPM fitting curve, the corresponding RPMs of the motor were calculated, when the impeller RPM was increasing from 0 to 200 RPM with a step of 25 RPM. The corresponding motor RPMs were about 0, 3030, 3770, 4360, 4810, 5180, 5500, 5800, and 6130. To characterize the linear property of our pump, the flow rate was measured in each corresponding RPM. Figure 7B shows the estimated RPM-the flow rate curve and the motor RPM-the flow rate curve. The flow rate was increased 0.037 times in inclination, from 0.93 mL/min at 25 RPM to 7.78 mL/min at 200 RPM. Those results are consistent with the pump affinity law in the centrifugal pump theory.

### 3.4. Comparison with Other Miniaturized Pumps

Many microfluidic devices such as the microblower-based pump and the acoutofluidic pump in Table 1 use the flow rate in the range of single microliters up to several hundred microliters per minute. However, for applications in analytical sample preparation, flow rates of a few milliliters per minute are sometimes used. The integrated microfluidic CD platform in Table 1 provided a flow rate range of 0.84 mL/min to 10 mL/min for multi-stepped biological and chemical assays. The minimum flow rate of our micropump, 0.54 mL/min, was higher than the flow rate range of micropumps suitable for sub-microscale fluidic manipulation devices, but it is proper to sample preparation steps.

The sizes of all pumps listed in the table are compact. However, when comparing in terms of the pump size, our pump is smaller than others. It looks that the microblower-based pump is smaller than ours, but it is the size of the commercial microblower itself and it cannot inject fluid into the microfluidic channel if the blower is not equipped with a horizontal liquid reservoir [20]. Therefore, the size of the microblower-based pump should be the sum of the microblower size and the reservoir size.

In addition, with regard to the pump operation, the listed pumps in the table require relatively large peripheral devices. All pump devices except ours required a DC power supply to apply the power of the pump systems. The centrifugal pump implanted in CD used a computer and controller software (Lab View) [22]. In the centrifugal pump fabricated using a 3D printer, a control circuit board to spin the rotor of the pump was needed [23]. Our pump requires only a motor system to rotate two external disk magnets and the size is only 95 mm × 70 mm × 75 mm. The motor rotates with a battery so that our entire micropump system is portable and available for POC tests.

With respect to material, other pumps are made of polymer; therefore, they have a limitation in the applications using organic solvents or acid/base because of the low chemical resistivity of the polymer material. On the other hand, our pump was made of glass, which has better chemical stability and higher mechanical strength than polymers. In addition, all elements were monolithically fabricated in a fused silica substrate using SLE. Thus, complicated assembly processes are eliminated, and the high durability of the built-in glass parts could be obtained.

## 4. Conclusions

We demonstrated a glass microfluidic centrifugal pump with integrated magnets. The glass microfluidic centrifugal pump system is composed of a glass pump microchip structure, a BLDC motor, and a motor controller. The dimension of the centrifugal pump was 11.4 mm × 9.72 mm × 2.96 mm. The 3D structures in the pump including the rotatable impeller were fabricated in a single glass using the SLE process. After two magnets were inserted into the impeller, it was covered with another glass sheet using a laser welding process. The minimum and maximum output pressure of the micropump depending on the motor RPM was 215 Pa at 2700 RPM and 3104 Pa at 6000 RPM. Regarding flow rate, its flow rate was linear in the range of 25–300 impeller RPM, and the minimum and maximum flow rates were 0.54 mL/min and 7.88 mL/min, respectively.

The glass micropump has the potential to be applicable in various fields. (1) The glass micropump can be used in various microfluidic devices using organic solvents and acid/base. It is fabricated using glass, especially fused silica, which exhibits high mechanical strength and chemical stability. Its properties make the glass micropump superior to organic polymers such as PDMS, which are mostly used to fabricate microfluidic devices. (2) The entire glass micropump system can be used for the on-site application. It measures only 95 mm × 70 mm × 75 mm including a battery so that it can operate without additional peripherals. (3) The micropump has highly durable glass structures. It is fabricated in a simple process that is SLE and laser welding. In the SLE process, all elements can be monolithically fabricated using one glass sheet, making in-glass parts durable. Then, the assembly of the micropump is sealed with reliable laser welding. With these advantages, the proposed glass micropump system can be used to provide an additional solution for lab-on-a-chips and point-of-care devices.

## Figures and Tables

**Figure 1 micromachines-13-01331-f001:**
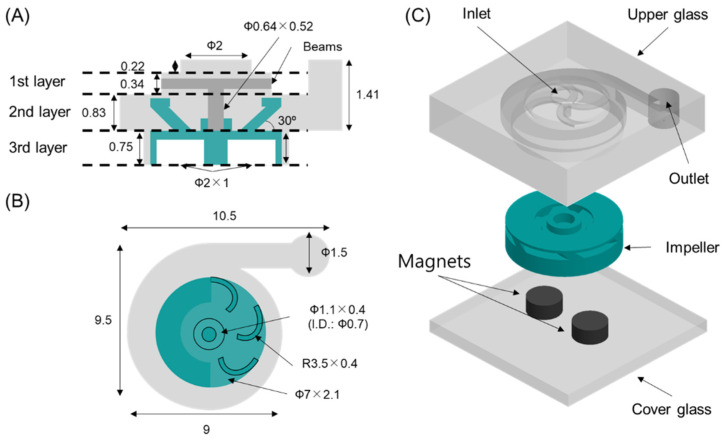
Schematics of a glass micropump. (**A**) side view (**B**) top view (**C**) 3D assembly (scale: mm).

**Figure 2 micromachines-13-01331-f002:**
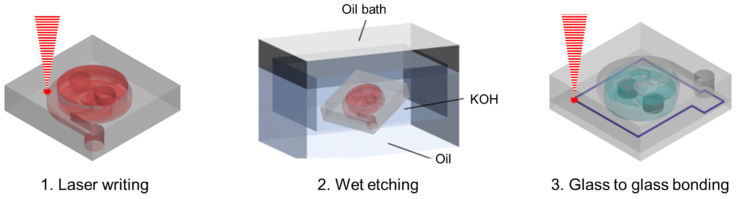
Fabrication process of the glass micropump.

**Figure 3 micromachines-13-01331-f003:**
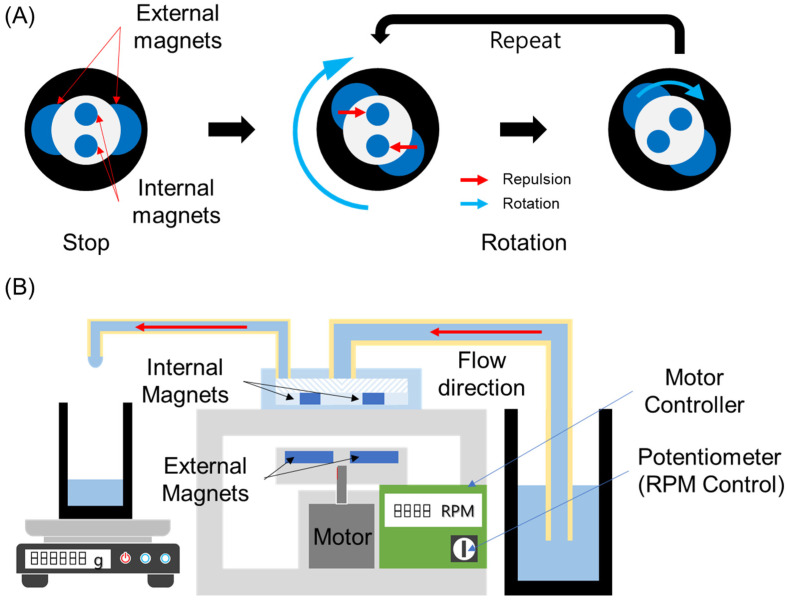
(**A**) An operation principle and (**B**) a performance test setup of the glass micropump.

**Figure 4 micromachines-13-01331-f004:**
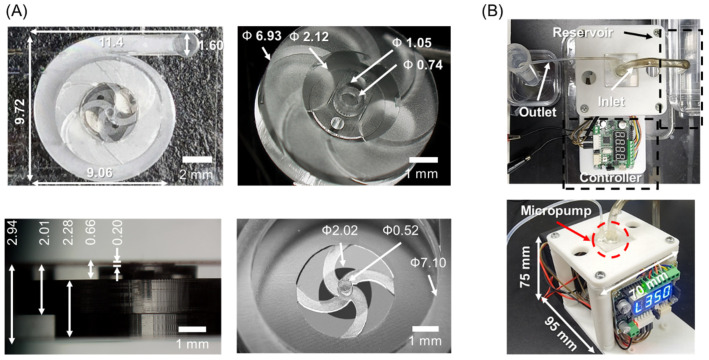
(**A**) Fabricated glass micropump and (**B**) performance test setup.

**Figure 5 micromachines-13-01331-f005:**
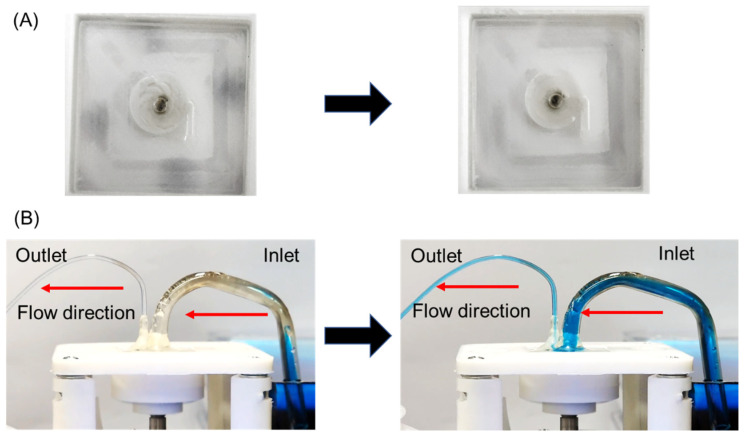
Operation results of the glass micropump. (**A**) Impeller rotation (**B**) Liquid suction and discharging.

**Figure 6 micromachines-13-01331-f006:**
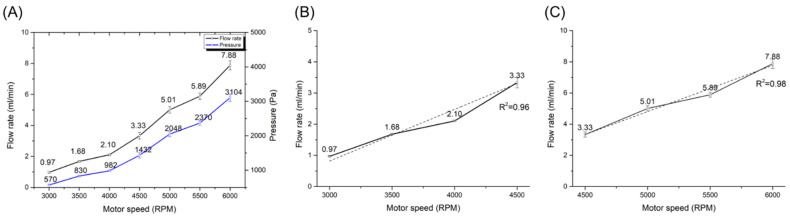
Pressure and flow rate curves of the glass micropump depending on the motor speed. (**A**) 3000–6000 RPM, (**B**) 3000–4500 RPM, (**C**) 4500–6000 RPM. In (**A**), blue and black lines are pressure and flow rate curves, respectively. In (**B**,**C**), the solid and the dotted line represent flow rate curve and flow rate trends, respectively.

**Figure 7 micromachines-13-01331-f007:**
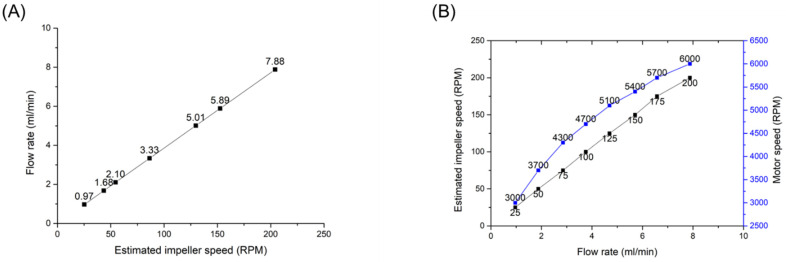
A flow rate curve according to an estimated impeller speed (**A**) and speed of the impeller and the motor according to the flow rate (**B**).

**Table 1 micromachines-13-01331-t001:** Comparison of miniaturized pumps with our device.

Pump Type	Q (mL∙min^−1^)	Pump Size (mm^3^)	Estimated Peripherals Size (mm)	Ref.
Acoustofluidics (3D-printed)	0.002–0.012	55 × 22 × 3	(1) Power supply (255 × 145 × 265) (2) Audio amplifier (Not shown)	[18]
Microblower-based	1 × 10^−10^– 128	20 × 20 × 1.85	(1) Reservoir (90 × 55 × 9) (2) Driver (30 × 40 × 5) (3) DC power supply (not shown)	[20]
Centrifugal (in CD)	0.84–10	25 (D) × 10 (estimated)	(1) Microfluidic CD with another motor (120 × 120 × 150) (2) Computer	[22]
Centrifugal (3D-printed)	103–124	28 × 30 × 24	(1) Control board (50 × 50 × 20) (2) DC power supply (213 × 89 × 348)	[23]
Centrifugal (SLE)	0.54–7.88	25 × 25 × 4	(1) Motor control system including a battery (95 × 70 × 75)	this work

## Data Availability

Not applicable.

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
