# Peer review of "A 3D Miniaturized Glass Magnetic-Active Centrifugal Micropump Fabricated by SLE Process and Laser Welding"

_micromachines, 2022, doi:10.3390/mi13081331_

Round 1

Reviewer 1 Report

The authors developed micro-pump for microfluidics. The flow range of the pump was 0.54 -7.88 mL/min, which is enough applicable for microfluidic experiment. The pump size was enough small for portable systems, which meets the concept of lab-on-a-chips and point-of-care devices in microfluidics. In addition, the pump fabricated by glass is great contribution to microfluidics users because organic solvent and acid/base also can be used as sample solutions. Based on these comments, the authors’ work will greatly contribute to microfluidics, therefore, the reviewer recommends the publication of this manuscript. However, the authors should address following comments.

1. In introduction part, not only microscale pumps but also nanoscale pumps can be introduced to enhance the explanation of the background of pumping technologies. The reviewer recommends some references for nanofluidic pumps: diffusio-osmotic pump (Phys. Rev. Lett. 112, 244501 (2014)), heat pump (RSC Adv., 7, 50591 (2017)) and etc.

2. For microfluidics users who use 100 um scale microchannels, 1 mL/min scale pumping is enough. However, in microfluidics, also many users who use 10 um scale microchannels expect 10-100 uL/min scale pumping. Therefore, the authors should also discuss the performance of the minimum flow rate by comparison with other groups in Table 1.

3. In microfluidics, some users use small and/or long microchannels which have huge hydrodynamic resistances. Therefore, the spec of the working pressure of the pump is important. The authors should show the performance of the pump with working pressure.

4. If possible, please show a movie about the rotation of glass micro-pump parts to help readers understand.

Reviewer 2 Report

The paper is written in a style that is difficult to understand.

Dimensional data are repeated without mentioning their influence compared to other pumps in the literature. The authors explain in the paper that they used glass (a type of glass used for optical devices such as diffraction gratings) to fabricate the magnetically driven pump because glass is chemically inert compared to polymers, but is not true.

The magnetic principle of operation of the pump is not described.

How does the glass and the refractive index of the glass affect the operation of the pump which is driven by magnetic forces induced by the external magnets and the magnets on the rotor which are of opposite polarity?

Some examples of sentences without logical meaning:

The abstract is incoherent (e.g. please rephrase: rows 14-17; 20-21). The authors use “in addition” too much, please consider changing it.

Row 115: how do you plan to maintain the velocity through the pump? Constant, increasing or decreasing?

Have you tried other dimensions of the inlet tube, besides the 8mm one? If yes, how does the flow rate modify when the tube is smaller, respectively larger than 8mm?

It is quite strange that the pump will not discharge fluid below 2700 rpm. And at 2700 the flow rate is 0.54 ml/min, which is a quite high flow rate for microfluidics. One suggestion is to change the microfluidic device, to a simpler one, with an aspect ratio of 1:1, in this way the pump may discharge below 2700 rpm and you can obtain smaller flow rates.

Row 189-192: “As we reported the elongation in depth while fabricating 3D structures in a glass sheet [32], all the heights of the glass micropump was compensated by about 1.47, the refractive index of  fused silica used in our process.”

Row 245: move coma after the equation

Volume rate does not have a physical meaning, consider changing it to volumetric flow rate, if you want to keep the volume part in the expression.

Reviewer 3 Report

The authors proposed a microfluidic centrifugal pump integrated with magnets. The glass microfluidic centrifugal pump system comprises a glass pump microchip structure, a BLDC motor, and a motor controller. The 3D structures in the pump including the rotatable impeller were fabricated in a single glass using the SLE process. The fabricating method is the laser welding process. 

1. In the introduction, the authors should investigate more types of miniaturized pumps. like, mechanical and non-mechanical pumps, etc.
for example, an Eccentric actuator driven by stacked electrohydrodynamic pumps. this kind of non-mechanical pump should be included. like other chemical pumps are also important. Autonomous oil flow is generated by self-oscillating polymer gels. 

2. A question about their design. I do not know why they use the magnet if they have to use the motor. Why not directly use the motor to drive the pumps? 

3. This pump's design and electrical parts are exciting and nice. 

4. Could you please explain why the sizes of the inlet and outlet (tube) are different? 

5. This will be very convincing if they make some simulations about the flow rate of their pumps. 

6. The comparison is nice. However, they should investigate the pressure performance of their pumps and the backpressure the pumps can withstand. Perhaps, a small example, like mixing the liquid also is necessary for demonstrating the excellent performance of their pumps. 

Round 2

Reviewer 1 Report

The authors well considered the reviewers' comments, and the manuscript was well revised based on the reviewers' comments. Therefore, the reviewer recommends the publication of this manuscript.

Author Response

Thank you for your time and effort to improve our manuscript, again. 

Reviewer 2 Report

The new revision of the paper has improved the quality of the work. Thus, the paper can be accepted.

Author Response

(The authors gave the same response as above.)

Reviewer 3 Report

I thank the authors for making such an excellent effort to incorporate the recommendations from both myself and the other reviewer. The quality of your responses to our comments is excellent. I feel the paper has gained a lot in terms of quality and clarity.

Small problem about comment 2 

I think the authors' device is remotely controlled rather than the leakage of fluids. The shaft-motor type is a normal pump in our daily life. 
